# Using Worker Position Data for Human-Driven Decision Support in Labour-Intensive Manufacturing

**DOI:** 10.3390/s23104928

**Published:** 2023-05-20

**Authors:** Ayse Aslan, Hanane El-Raoui, Jack Hanson, Gokula Vasantha, John Quigley, Jonathan Corney, Andrew Sherlock

**Affiliations:** 1The School of Computing, Engineering and The Built Environment, Edinburgh Napier University, Edinburgh EH10 5DT, UK; 2Strathclyde Business School, University of Strathclyde, Glasgow G1 1XQ, UK; 3School of Engineering, The University of Edinburgh, Edinburgh, EH8 9YL, UK; 4National Manufacturing Institute Scotland, Glasgow PA3 2EF, UK

**Keywords:** industrial productivity, process mining, discrete event simulation, indoor positioning systems, completion time, flexible capacity allocation

## Abstract

This paper provides a novel methodology for human-driven decision support for capacity allocation in labour-intensive manufacturing systems. In such systems (where output depends solely on human labour) it is essential that any changes aimed at improving productivity are informed by the workers’ actual working practices, rather than attempting to implement strategies based on an idealised representation of a theoretical production process. This paper reports how worker position data (obtained by localisation sensors) can be used as input to process mining algorithms to generate a data-driven process model to understand how manufacturing tasks are actually performed and how this model can then be used to build a discrete event simulation to investigate the performance of capacity allocation adjustments made to the original working practice observed in the data. The proposed methodology is demonstrated using a real-world dataset generated by a manual assembly line involving six workers performing six manufacturing tasks. It is found that, with small capacity adjustments, one can reduce the completion time by 7% (i.e., without requiring any additional workers), and with an additional worker a 16% reduction in completion time can be achieved by increasing the capacity of the bottleneck tasks which take relatively longer time than others.

## 1. Introduction

Labour-intensive manufacturing systems [1], like workshops, often experience deviations from planned production schedules due to human workers’ unpredictable and flexible behaviour. This behaviour differs from automated machines, which perform tasks as prescribed. Deviations can occur even with fixed task assignments due to workers’ abilities and conduct. For example, workers can interrupt their own task and help colleagues with tasks at other work stations when they observe that their coworkers are having difficulties. Understanding when and why workers demonstrate flexible working practices can help manufacturers learn and adjust their decision-making processes (e.g., re-planning the task assignment) accordingly, leading to successful and worker-accepted process improvements. Such human-driven strategies are more likely to succeed compared to those based on idealised performance.

In such industrial environments, sensor technologies are key to understanding worker behaviour and how they actually perform manufacturing tasks. Since in labour-intensive manufacturing systems human workers are the backbone of the production process, to understand how specific manufacturing tasks are actually being performed it is necessary to track worker movements using indoor positioning systems, such as Ultra-wideband tags, which are becoming increasingly common in production facilities [2]. Real-time position data from these sensors can be used to analyse how tasks are executed by workers in labour-intensive manufacturing processes. Our paper demonstrates this using a real-world dataset [3] of six assembly workers with an initial task assignment.

Real-time position data from workers requires analysis and data mining techniques to derive a meaningful representation of how workers perform tasks. This paper proposes a process mining [4] methodology to derive the sequence of assembly operations carried out by human workers performing a manufacturing task and use this representation to define a process model. By comparing the models derived from the human worker data to their nominal (i.e., fixed) task assignment, deviations are identified that workers are sometimes involved in tasks that are not assigned to them as a result of their adaptive and collaborative behaviour. Through this comparison, it is discovered that workers exhibit more flexibility than their assigned tasks suggest, and are capable of performing a variety of tasks and distributing their capacity among them. Using this flexible task assignment, we provide decision support for improving process efficiency and reducing completion time by making minor adjustments to task capacity allocation. Previous studies [5,6,7] have highlighted manpower allocation as a critical aspect of decision-making in labour-intensive manufacturing. However, most of these studies rely on analytical models that make strict assumptions about how workers should perform tasks. To evaluate the impact of proposed changes to labour allocation, this paper builds a discrete event simulation using the process model generated from worker localisation data, allowing us to assess completion time under new capacity levels. This paper extends the initial results presented in conference paper [8] by demonstrating the potential of worker position data for providing human-driven decision support via data-driven process models and simulation tools.

The rest of this paper is organised as follows. In Section 2, we review the literature and show how this paper is different from the relevant existing studies. Section 3 describes our methodology and outlines its steps. Section 4 presents our main results. Lastly, Section 5 concludes the paper while pointing to several future research directions.

## 2. Literature Review

Real-time localisation of objects in manufacturing environments, such as people, machines, materials, and products is essential to achieve the objectives of smart manufacturing and Industry 4.0 revolution which depend on the integration of knowledge about the production environment continuously [9,10]. The specific real-time localisation system (RTLS) technologies that enable this in manufacturing are the indoor positioning systems (IPS). Based on the communication technology used, these systems can be grouped into Wi-Fi, Bluetooth, radio frequency identification (RFID), VLC, and ultra-wide band (UWB)-type technologies [11]. The position data supplied by these technologies have the potential to be used for monitoring and control of manufacturing processes in order to improve efficiency, detect faults or anomalies [12]. For example, tagging and tracking unfinished products provide information, such as tact time, which enables efficiency monitoring and production control [13].

Table 1 summarises the previous studies that made use of indoor localisation sensor data for decision support in manufacturing.

The literature summary in Table 1 suggests that the majority of the studies focus on tracking products to monitor efficiency and provide decision support for process improvement. We note the works by Nwakanma et al. [20] and Islam et al. [22] who also track workers inside the manufacturing environments, as in our methodology. In these papers, authors focus on monitoring safety and detecting dangerous situations. Their approach also involves integrating data on breathing patterns of workers. In contrast, our paper uses worker position data in conjunction with the factory plan to assess and improve process efficiency.

In Table 1, the work of [14,15,16] should be acknowledged, who, like us, utilise simulations for decision support. However, a key distinction between our approach and theirs pertains to the modelling of worker behaviour during manufacturing tasks in the simulation. Our paper adopts a data-driven approach that leverages worker position data and process mining techniques. In contrast, these studies rely on manual inputs, such as expert opinions.

Process mining algorithms automatically discover, monitor, and improve actual as-is processes (i.e., not assumed processes) from event logs, with applications in process analysis, improvement, compliance, and case management. In manufacturing, it is the third largest application domain and is used to identify process deviations and detect bottlenecks, typically relying on event logs generated by IT tracking systems of activities performed on specific products [4]. For instance, [24] uses an order status tracking system to create event logs for ordered products and the activities performed on them, while [25] relies on code readers placed on machines to automatically scan product IDs and record timestamps. [26] also includes product information. Perhaps, the most relevant study to our paper is by Tran et al. [21], which we note in Table 1, who also uses position data from localisation sensors to extract event logs for process mining. However, in all these studies, the event logs are based on traces of products, whereas our methodology is based on tracking workers.

Product positions can provide a more direct understanding of activities compared to worker positions, but detecting manufacturing activities from worker positions requires activity recognition. Combining process mining with activity recognition is crucial for manufacturing, as noted in [27]. Our methodology uses the facility layout to detect activities related to manufacturing tasks using worker position data, similar to [19,21]. A relevant study in the context of tracking worker movement for process mining to discover process models in manufacturing is [28]. In [28], a video-based system tracks hand movements of workers in a manual assembly line to identify activities and create event logs. Similarly, our paper uses process mining to discover process models and identify deviations in worker behaviour. Additionally, our paper uses the data-driven process model to provide decision support on capacity allocation through discrete event simulation.

Discrete event simulation (DES) models provide decision support for complex systems by allowing what-if analyses and exploration of process redesign alternatives [29]. Traditionally, DES models are manually built using expert knowledge, but data-driven models offer advantages [30]. Process mining, which leverages event logs from actual process executions, can be particularly valuable in developing data-driven DES models [31,32]. In this approach, process models and performance information derived from process mining can serve as the input for automating the design of simulation models. While previous studies have focused on healthcare [33,34], applications in manufacturing are emerging [30].

The focus of this paper is to provide human-driven decision support to improve process efficiency. However, we would like to also touch upon the link between the human-centric decision-making and sustainability [35] and resiliency-related objectives [36]. We note that in manufacturing systems in particular these aspects become important to aim through human-centricity in human–robot collaboration systems [37].

## 3. Materials and Methods

### 3.1. Data

This paper uses the 2D worker position data provided in [3] which are collected through indoor localisation sensors during a three-hours long work shift. The dataset is generated using both UWB tags worn by workers and motion capture technologies to track workers. Here, we use their UWB data because the motion capture system provides fewer position samples (since it was used for only about two hours). Specifically, these data provide the position measurements taken from each worker in x and y coordinates. These coordinates are measured in meters by considering the anchor position (0, 0) as the reference point on the shop floor. Figure 1 presents a scatter plot of the 2D position data coming from one of the workers for visualisation purposes.

The manufacturing process that workers are involved is an assembly line of tricycles. This line consists of six different manufacturing tasks (j=1,2,…,6) that are to be performed by six workers (i=1,2,…,6). Figure 2 presents the tasks and the precedence relations between them (e.g., the axle cannot be assembled without having the lower frame).

In this assembly line, each worker is fixed to a single task. So, the fixed task assignment can be described with the following matrix given in Equation (Equation 1).
(1)$$A^{\mathrm{fixed}}=\left[a_{ij}^{\mathrm{fixed}}\right]=\begin{array}{cc} & \begin{array}{cccccc} Task\;1 & Task\;2 & Task\;3 & Task\;4 & Task\;5 & Task\;6 \end{array}\\ \begin{array}{l} Worker\;1\\ Worker\;2\\ Worker\;3\\ Worker\;4\\ Worker\;5\\ Worker\;6 \end{array} & \begin{array}{ccc} ( & \begin{array}{cccccc} 1 & \;\;\;\;\;\;\;\;\;\;0 & \;\;\;\;\;\;\;\;\;\;0 & \;\;\;\;\;\;\;\;\;\;0 & \;\;\;\;\;\;\;\;\;\;0 & \;\;\;\;\;\;\;\;\;\;0\\ 0 & \;\;\;\;\;\;\;\;\;\;1 & \;\;\;\;\;\;\;\;\;\;0 & \;\;\;\;\;\;\;\;\;\;0 & \;\;\;\;\;\;\;\;\;\;0 & \;\;\;\;\;\;\;\;\;\;0\\ 0 & \;\;\;\;\;\;\;\;\;\;0 & \;\;\;\;\;\;\;\;\;\;1 & \;\;\;\;\;\;\;\;\;\;0 & \;\;\;\;\;\;\;\;\;\;0 & \;\;\;\;\;\;\;\;\;\;0\\ 0 & \;\;\;\;\;\;\;\;\;\;0 & \;\;\;\;\;\;\;\;\;\;0 & \;\;\;\;\;\;\;\;\;\;1 & \;\;\;\;\;\;\;\;\;\;0 & \;\;\;\;\;\;\;\;\;\;0\\ 0 & \;\;\;\;\;\;\;\;\;\;0 & \;\;\;\;\;\;\;\;\;\;0 & \;\;\;\;\;\;\;\;\;\;0 & \;\;\;\;\;\;\;\;\;\;1 & \;\;\;\;\;\;\;\;\;\;0\\ 0 & \;\;\;\;\;\;\;\;\;\;0 & \;\;\;\;\;\;\;\;\;\;0 & \;\;\;\;\;\;\;\;\;\;0 & \;\;\;\;\;\;\;\;\;\;0 & \;\;\;\;\;\;\;\;\;\;1 \end{array} & ) \end{array} \end{array}$$

The manufacturing floor is divided into regions and each task has its own designated region. The locations of these regions are shown in Figure 3.

### 3.2. Methodology

Given this floor plan and the worker position data collected through UWB sensors, we develop our methodology to extract process models and incorporate them in a discrete event simulation framework to provide decision support. This methodology is summarised in Figure 4.

Our methodology involves a number of steps. These are described in detail below.

#### 3.2.1. Data Processing and The Extraction of an Event Log

UWB tags are measured at an interval of 100 ms. To align the data collected from UWB tags attached to workers, the data interval of 100 ms is converted to seconds as no significant changes in worker positions are expected within a second. The average position data obtained within the same second is used as a smoothing function. If data are missing for a worker, for example, when signals are obstructed, this is substituted with the worker’s last known position.

An event log is generated from the synchronised second-by-second position dataset. To assign events relating to the execution of a certain manufacturing task by workers, we examine the proximity of workers to task regions and the duration of their stay there. To identify whether a worker is actually performing a specific task based on their position, the amount of time they spend in a particular region should be considered. For example, if a worker is in one region at second *t* and then in another region the next second, it could simply be due to walking past, particularly if their time in the initial region was brief. To address this, a parameter τ>0 is used to determine if a worker is performing a task *i* at some time *t* if their position was within task region *j* at any time $t^{{}^{\prime }}$ within the period t,t−1,⋯,t−τ. Here, by fixing τ to 60 s, we determine the log of events that show the execution of a certain task by a certain worker with a start and end time. It must be noted that τ is not set to a lower value here to avoid considering short visits to the regions of other tasks (e.g., transfer of products and material) as workers being involved in the execution of these tasks.

#### 3.2.2. Process Mining to Derive Process Models

In this step, the event log is used to discover a process model of the manufacturing process with a suitable process mining algorithm. Multiple process model discovery algorithms, including Alpha Miner, Heuristic Miner, and Inductive Miner, are available for discovering process models from event logs. Similar to [38], this paper applies the Inductive Miner, since it is an improvement on the Alpha and Heuristic algorithms. The Inductive Miner is a divide and conquer type polynomial-time algorithm that returns a process tree in which each activity is seen once. We refer the reader to [39] for the theoretical foundations of this algorithm.

This paper uses the implementation of this algorithm available in ProM 6.11 (https://www.promtools.org/, accessed on 20 March 2023). The event log is provided as a comma-separated file to ProM, as illustrated in Figure 5. Before the Inductive Miner is applied, the converter tool in ProM is used to to convert this to XES format.

#### 3.2.3. Building the Discrete Event Simulation

This paper builds a simulation model based on the process model obtained from the event log. The process model obtained from the Inductive Miner algorithm uses the business process model notation (BPMN) [40]. This notation can consist of objects that mark specific events, including the start and end of the process, which are denoted with small circles; activities, which are represented with rectangles; connections and flow objects showing the sequence relations, which are represented using arrows; and also gateways, which are shown with smaller diamond shapes. Gateways are objects that determine forking and merging of paths and they can model conditional events. For example, parallel gateways (e.g., splitting or joining) are used to model the completion of parallel processes that can only finish when each of its processes is performed.

To build a discrete event simulation framework that represents the state of the process and events that transform the state under the transition probabilities, this BPMN notation should be translated. Our approach models the start, end and intermediate events, and also activities as part of the state description and uses flow sequences and their frequencies to model the transition probabilities. We call the model of the entire process as $Process^{\mathrm{main}}$ which consists of several subprocesses because of the parallel gateways in the process model. The main process always starts from the same state, the start event marker in the process model, and will end the first time the state transitions to the end state. To complete the simulation framework the transition times should also be modelled, namely, how long it will take until the current state *s* will change to another state $s^{\prime }$ due to the completion of a task execution event. To model the transition times, we use the mean sojourn time information as supplied by the Inductive Miner and use stochastic distributions whose means are fitted to these. The details of how this simulation framework is built with the data of our case study are given in the Appendix A. We implement this simulation in C++ language, using Visual Studio.

The performance measure evaluated with this simulation is the average completion time of the manufacturing process of assembling 6 tricycles. In this evaluation, a large number of samples (K=50,000) (to reduce the standard error) are taken, from multiple simulation runs under random seeding, and then the averages of their completion times are calculated. The completion time of the process from each sample *k* is the first passage time until $Process^{\mathrm{main}}$ reaches the end state. We denote the completion time with $C(Process^{\mathrm{main}})$. So, based on the simulation samples, this is measured with
(2)$$C\left(Process^{\mathrm{main}}\right)=\frac{1}{K}\sum_{k=1}^{K}C\left(Process_{k}^{\mathrm{main}}\right),$$
where $C(Process_{k}^{\mathrm{main}})$ is the completion time measured in the *k*th sample.

#### 3.2.4. Using the Simulation for Decision Support on Capacity Allocation

To facilitate the use of the simulation model for the allocation of workers to tasks, it is necessary to establish the relationship between the labour resource (e.g., person-minutes) and output for each stage of the process. This paper achieves this by fitting the total capacity allocated to the tasks, as observed from the event log, to the mean sojourn times of tasks, as provided in the process model derived with the Inductive Miner. Then, considering that with more capacity allocated to a task the faster its execution will be, we find the corresponding new mean task sojourn times to be used in the simulation model under a new capacity allocation. A linear model is used for this purpose.

The first step is the derivation of the task assignment matrix of the data ($A^{\mathrm{data}}=\left[a_{ij}^{\mathrm{data}}\right]$) based on the event log. This is obtained from the total time that workers spend for executing each of the manufacturing tasks. Then, from this the total capacity allocated to each task in the process observed with the data $R_{j}^{\mathrm{data}},\forall j=1,2,\ldots ,6$ is found via
(3)$$R_{j}=\sum_{i}a_{ij}^{\mathrm{data}}.$$

Letting $\mu_{j}^{\mathrm{mined}}$ denote the mean sojourn time of task *j* as obtained from the Inductive Miner, and $\mu_{j}^{\mathrm{new}}$ the new (would be) mean sojourn time of task *j* under $R_{j}^{\mathrm{new}}$, a new total capacity level for task *j*, we have
(4)$$\mu_{j}^{\mathrm{new}}=\frac{\mu_{j}^{\mathrm{mined}}R_{j}^{\mathrm{data}}}{R_{j}^{\mathrm{new}}}.$$

So, our decision support involves changing the task allocation under a new task allocation matrix $A^{\mathrm{new}}$ and having new capacity levels for the tasks $R_{j}^{\mathrm{new}},\forall j=1,2,\ldots ,6$, updating the mean sojourn times of the tasks (the transition times in the simulation) in the simulation and using the simulation to evaluate the average completion time under the new allocation. The aim of this procedure is to find suitable allocations that will reduce the completion time.

## 4. Results

### 4.1. Observing Deviations from the Fixed Task Assignment and Finding the Assignment from the Data

This section explores the event log to observe and identify deviations from the fixed task assignment, as given in Equation (Equation 1). For this, using the dotted chart visualisation tool in ProM first, Figure 6 is presented. The first observation to make is that the supposed task assignment is not being strictly followed, as there are deviations. Specifically, it is observed that worker 2 joins worker 1 for task 1 at the beginning of the shift, and worker 4 sometimes takes on the responsibility of task 5. Additionally, worker 6 assists with tasks 2 and 3. These deviations may be due to the fact that the tasks assigned to workers 2, 4, and 6 are dependent on outputs from earlier assembly stages and cannot be started independently. Additionally, it can be seen that some workers perform task continuously throughout the shift, such as worker 5. However, we typically see that the execution of tasks are clustered and there are breaks in between. These breaks may be due to the completion of the current product and switching to another, since a total of six tricycles are completed during the shift.

Secondly, from the event log, the task assignment matrix of the data ($A^{\mathrm{data}}=\left[a_{ij}^{\mathrm{data}}\right]$) based on how much time each worker spends on performing each of the manufacturing tasks is calculated. This matrix is presented in Equation (Equation 5), which again demonstrates how workers deviate from the fixed task assignment and how much of their working capacity is put into several tasks. According to this matrix, it is found that $R_{j}^{\mathrm{data}},j=1,2...,6$, the total capacity put to each task according to these data as (1.28,1.21,1.41,0.55,1.32,0.24), which indicates flexible worker behaviour. This shows that tasks 1, 2, 3, and 5 are given similar capacity levels, while the capacity allocated to tasks 4 and 6 is much lower. Considering what these tasks involve (see Figure 2), we see that tasks 4 and 6 are characteristically similar and somewhat different from other tasks such that they involve some final assembly steps of a number of sub-assemblies. According to $A^{\mathrm{data}}$, it seems that workers collectively put less effort into these assembly steps.
(5)$$A^{\mathrm{data}}=\left[a_{ij}^{\mathrm{data}}\right]=\begin{array}{cc} & \begin{array}{cccccc} Task\;1\;\; & Task\;2\;\; & \;\;\;\;\;Task\;3 & \;\;\;\;\;Task\;4\; & \;\;\;Task\;5 & Task\;6\;\; \end{array}\\ \begin{array}{l} Worker\;1\\ Worker\;2\\ Worker\;3\\ Worker\;4\\ Worker\;5\\ Worker\;6 \end{array} & \begin{array}{ccc} ( & \begin{array}{cccccc} 1 & \;\;\;\;\;\;\;\;0 & \;\;\;\;\;\;\;\;\;\;0 & \;\;\;\;\;\;\;\;\;\;0 & \;\;\;\;\;\;\;\;\;\;0 & \;\;\;0\\ 0.28 & \;\;\;\;\;0.31 & \;\;\;\;\;\;\;\;\;\;0 & \;\;\;\;\;\;\;\;\;\;0 & \;\;\;\;\;\;\;\;\;\;0.24 & \;\;\;0.18\\ 0 & \;\;\;\;\;0 & \;\;\;\;\;\;\;\;\;\;1 & \;\;\;\;\;\;\;\;\;\;0 & \;\;\;\;\;\;\;\;\;\;\;0 & \;\;\;\;\;0\\ 0 & \;\;\;\;\;0 & \;\;\;\;\;\;\;\;\;\;0 & \;\;\;\;\;\;\;\;\;\;0.55 & \;\;\;\;\;\;\;\;\;\;0.45 & \;\;\;\;\;0\\ 0 & \;\;\;\;\;0 & \;\;\;\;\;\;\;\;\;\;0.37 & \;\;\;\;\;\;\;\;\;\;0 & \;\;\;\;\;\;\;\;\;\;0.63 & \;\;\;\;\;0\\ 0 & \;\;\;\;\;0.90 & \;\;\;\;\;\;\;\;\;\;0.04 & \;\;\;\;\;\;\;\;\;\;0 & \;\;\;\;\;\;\;\;\;\;0 & \;\;\;\;0.06 \end{array} & ) \end{array} \end{array}$$

### 4.2. Data-Driven Process Model

It is possible to obtain a variety of different process models using the Inductive Miner. This is made possible by varying the two filtering parameters of the algorithm. These are the filters on the activities, namely, the manufacturing tasks, and the paths among activities as observed in the event log. Varying the level of filtering applied affects the quality of the obtained process models with respect to their fitness and complexity. Model fitness, which measures the ability of the model to allow the behaviour observed in the data (i.e., the ability to replay the sequences observed in the logs), and simplicity are among of the most commonly considered process model quality metrics along with the quality measures of precision (i.e., not allowing the behaviour not seen in the logs) and generalisation. We refer the reader to [4] for more information on these four quality measures.

Specifically, with increased levels of filtering, simpler models can be obtained, however, their fitness may be lower. Our paper applies 10% filtering on the paths and obtain a model with 96% fitness (Figure 7). Here, we do not use the filtering on the activities so that we can see the involvement of each manufacturing task in the process. This is preferred because this model is to be used for building a discrete event simulation which will be used for the capacity allocation decisions to each of the manufacturing tasks. It can be observed that the model includes processes that are executed in parallel and forwarded to the next activity when all the involved processes are completed, as indicated with the splitting and joining parallel gateways. This dependency can be as a result of the precedence relations between the manufacturing tasks, as illustrated in Figure 2.

The model in Figure 7 is used when fitting our discrete event simulation. The mean sojourn times of the tasks $\mu_{j}^{\mathrm{mined}}$ are (3.07,4.18,2.20,22.15,3.06,3.59) in minutes. Although this suggests that the execution of task 4 takes the longest, it must be noted that this does not mean that the total time spent for task 4 until the process ends will be the longest. This is because the total time spent on tasks also depends on how many times they will have to be executed, information which is captured in the flow frequencies in the process model. It can be seen in Figure 7 that the number of times that task 4 will be executed is much lower than other tasks. For this reason, the flow frequencies must be accounted for, in our discrete event simulation. This paper investigates the total time spent in tasks using our simulation.

### 4.3. Decision Support on Capacity Allocation with Simulation

This section conducts a series of investigations for the purposes of decision support regarding the capacity allocation. In doing so, we consider new task assignments and capacity allocation levels and use our discrete event simulation to evaluate the average completion time. In every evaluation, *K* = 50,000 samples are taken from the simulation, as described in Section 3.2.3. This gives a high-precision estimate on the average completion time with standard errors falling below 1.4 min the most of the time. Nevertheless, the errors are provided in our results. The time unit used in the results presented in this section is minutes.

#### 4.3.1. Comparing the Performance under the Worker-Driven Flexible Allocation to the Fixed Assignment

This section firstly evaluates the completion time according to the flexible allocation observed in the worker data (as manifested in $A^{\mathrm{data}}$ and $R_{j}^{\mathrm{data}}$). Secondly, the completion time according to $A^{\mathrm{fixed}}$, namely, under $R_{j}^{\mathrm{fixed}}=1,\forall j$ is evaluated. In other words, the second evaluation measures the completion time if the workers stuck to the fixed assignment prescribed to them and did not become involve in any other tasks besides the ones that were assigned to them. Comparing the completion time under these two allocation schemes is useful to understand if the flexible worker behaviour as observed in the data (and the practice of assisting with the other manufacturing tasks when necessary) resulted in an improved performance (i.e., completion of the day’s workload earlier).

The average completion time under the fixed allocation gives 328.6±1.4, but this reduces to 309.8±1.4 using the worker-driven flexible allocation. Considering the error margins, the difference in completion times is significant and indicates an improvement around 5.7% in the average completion time under the flexible allocation compared to the fixed allocation. This shows that indeed the workers’ flexible behaviour benefits the overall efficiency of the entire manufacturing process and is a more appropriate approach than the strict allocation of fixed assignment.

Figure 8 presents the box and whisker plots of the completion times under the fixed and flexible allocations to illustrate the distributions of the completion times in more detail. We find that the medians of completion times are 211 and 222 min under the flexible and fixed allocations, respectively. Note that these values are close to the original total duration of the work shift observed in the data, which was around 180 min (3 h).

The total capacity allocated to each task in the worker-driven flexible allocation $R^{\mathrm{data}}=\left(1.28,1.21,1.41,0.55,1.32,0.24\right)$ is significantly different than the fixed allocation $R^{\mathrm{fixed}}=\left(1,1,1,1,1,1\right)$. For example, more capacity is allotted to the tasks 1–3 and 5 under the flexible allocation, which, at the same time, results in a lower capacity for the tasks 4 and 6 compared to the fixed allocation. Because of this, the time spent for performing the specific manufacturing tasks in completing a day’s work would be different as well. Figure 9 presents these task-specific times. It can be observed that how the increased capacity for the tasks 1–3 and 5 in the flexible allocation decreases the time spent in completing these tasks, while it increases the time required for the tasks 4 and 6. In addition, another important difference is noted in the times spent for tasks under these two capacity allocation schemes, which is about the variance observed in the times across all six tasks. We calculate the standard deviation of the average of the total times across tasks as 59.9 and 38.3 in the fixed and flexible allocations, respectively. This reveals that the times spent in tasks are more balanced under the flexible allocation. It can be argued that this balancing advantage over the fixed assignment could be a contributing factor to the lower completion time achieved with the flexible allocation, since the completion of the main manufacturing process depends on parallel subprocesses. More specifically, the completion time of a parallel process involving several tasks highly depends on its bottleneck (i.e., slowest) task since the parallel process can finish only when all of the tasks are finished and a balanced timing of the tasks can help to reduce the time required for this completion.

#### 4.3.2. How to Reduce Completion Time through Small Adjustments Based on the Worker-Driven Flexible Allocation?

The aim of this section is to provide insights to workers based on their natural work practices, as demonstrated in the flexible allocation $R^{\mathrm{data}}$, to ease their workload and finish their shifts earlier. We achieve this by considering small adjustments on the flexible capacity allocation $R^{\mathrm{data}}$, evaluating the completion time under the resulting new allocation scheme, and investigate whether it is possible to reduce the completion time of their work through this re-allocation of their capacity. In these evaluations, capacity adjustments that reduce the capacity from some task *j* by some percentage and transfer this to some other task $j{}^{\prime }$ are considered. The completion times under the evaluated schemes that shift 5%, 10%, 20%, and 50% capacity from one task to another are given in Table 2. Larger capacity shifts are not shown in Table 2 as no improvements in the completion time are found.

Given that the original un-adjusted scheme gives an average completion time of 309.8±1.4, we see that there are adjustments that can reduce the completion time significantly. The best adjustments that lead to the highest reductions are identified in bold in Table 2. These adjustments, which suggest shifting capacity from task 1 to task 4, from task 3 to task 4, from task 3 to task 5, and from task 3 to task 6, are able to achieve a reduction between 3 and 5%. Considering the total time spent in each task (see Figure 9), it is observed that each of these moves shifts capacity from a task taking shorter time to a task taking a relatively longer time. For example, it is known that task 3 is the task taking the shortest amount of time in the current assignment and this investigation finds several re-allocations to shift capacity from this task to other tasks that take longer, such as the tasks 4, 5, and 6. In other words, these capacity shifts are in favour of balancing the time required to complete each one of the manufacturing tasks. The reason that they are able to reduce the completion time could be due to this balancing effect since the process includes parallel tasks and they are dependent on each other. When the amount of capacity shifted is considered in these adjustments, it is found that the best reductions are possible through smaller but considerable shifts of 20%, whereas higher shifts (50%) start to lose their advantage in reducing the average completion time. Note that large capacity shifts mean large reductions in capacity from the tasks and this may cause delays in the process. For example, even when the task that the capacity is shifted to is a bottleneck and needs capacity, a large reduction from the capacity of another task to fix this may simply shift the place of the bottleneck without achieving significant reduction in the overall completion time. In the last row of Table 2, we investigate a combination of the two of the best small adjustments, shifting 20% capacity of task 1 to task 4 and 20% capacity of task 3 to task 6. With this, the reduction in completion time reaches above 7%, making it possible to finish their work around 20 min earlier. This shows that these small adjustments have potential to make considerable reductions in the completion time when they are based on the workers’ behaviour as demonstrated in their movement data.

#### 4.3.3. How to Use Additional Worker Capacity Based on the Worker-Driven Flexible Allocation?

The previous investigation provides insights for which small adjustments to the worker-driven flexible allocation improves the completion time best such that the workers can complete their work earlier, without requiring to hire and engage additional worker capacity. The assembly system we analyse in this paper has, in total, six workers. This section investigates which capacity allocation strategy would be the best if there was one more worker available to assist in performing the tasks. For this, the worker-driven flexible allocation $R^{\mathrm{data}}=\left(1.28,1.21,1.41,0.55,1.32,0.24\right)$ is considered as the current practice under the capacity of six workers and then allocate the available one unit capacity of the additional worker to improve this.

Firstly, the strategies that concentrate the entire additional capacity on one single manufacturing task are evaluated (Figure 10). This shows that the best strategy is to assign the new worker to task 5 and the second best option would be task 4. With the strategy of assigning the entire additional capacity to task 4, the completion time reduces from 309.8±1.4 to 273.3±1.2, having a reduction potential around 11.6% in the completion time. As discovered in the previous investigations, this preference for the tasks 4 and 5 seem to be in line with the logic of trying to balance the time spent on the manufacturing tasks. From Figure 9, it is known that these tasks take the largest amount of time according to the current practice $R^{\mathrm{data}}$ with six workers and assigning the new worker to these would have a balancing effect on the time spent on each one of the tasks by reducing the time for these tasks.

As demonstrated in the data, it is known that workers can flexibly allocate their capacity and become involved in multiple tasks during their shift. Keeping this in mind, flexible strategies that assigns the new worker to more than one task are investigated next. Figure 11 considers strategies that split the capacity in two in order to support two different manufacturing tasks. It can be observed that the strategy that assigns the half of the new worker’s capacity to task 4 and the other half to task 5 is the best. Moreover, it seems that this split strategy reduces the completion time much better than the single-task assignment strategies of assigning the entire capacity to task 4 or task 5, achieving a reduction of around 16.8%. Even more flexible allocations that would assign the new worker to more than two tasks have the potential to reduce the completion time more. However, this can create a complicated situation for the new worker in terms of scheduling and time management. Moreover, because of these challenges it may not be possible to realise the high-reduction potential of such multi-task assignment strategies are beyond the scope of the current work.

## 5. Conclusions and Discussion

This paper presents a novel methodology for supporting human-driven decision-making in labour-intensive manufacturing processes by analysing worker position data. This involves integrating process mining and discrete event simulation to identify opportunities to improve capacity allocation decisions. Our main goal is to demonstrate the value of using localisation sensor data for data-driven decision-making in manufacturing, with a focus on improving process efficiency. By showcasing the benefits of this technology, our aim is to help manufacturers make informed decisions about whether to invest in indoor localisation sensors or not.

This methodology is applied to a real-world dataset involving six workers performing six manual manufacturing tasks to assemble tricycles with a fixed assignment in which each worker is responsible for performing one of the tasks. Our first identification is that workers demonstrate a much more flexible behaviour than the fixed task assignment and involve in various tasks. By putting this flexible assignment observed in the data in the centre, this paper provides decision support on how to reduce the completion time of the manufacturing process through small adjustments on the capacity levels that workers allocated to the tasks. In our capacity adjustment investigations, we first show that with small adjustments of shifting capacity between two manufacturing tasks one can reduce the completion time by 7%, without requiring any additional workers. Secondly, we investigate the situation where there is one more worker available to support the manufacturing process and show that this additional capacity can reduce the completion time by more than 11% if the new worker is to be assigned to a single task, while this can reach above 16% if two tasks are assigned.

Our work presented in this paper has certain limitations. Firstly, our approach for detecting specific manufacturing tasks performed by workers may face challenges with accuracy. This is because our approach relies on position data provided by sensors and the factory layout, which dictates the regions where workers can perform each task. For instance, inaccurate sensor data or workers performing tasks outside their designated zones may result in inaccuracies or failures in detecting activity events. Another limitation of our research is the allocation of worker capacity. While suggesting adjustments to the capacity levels allocated to tasks, we only consider the total available capacity of workers. This may overlook scheduling decisions and inefficiencies that may arise due to complications in assigning workers to multiple manufacturing tasks.

Future research can delve into decision support on dynamic capacity allocation and/or worker assignment strategies. This paper used the entire data coming from a work shift to find a process model representing the process from beginning to end. However, to understand the changing work patterns in time, the data can be split into several segments in time and, instead of deriving a single process model, one can obtain multiple models with process mining, by considering the event logs of each segment separately. In this way, one can build models of the process segmented in time and investigate dynamic strategies that can vary in time. For example, it could be that allocating more capacity to the first task is beneficial during the first hour of the shift but not so in the last hour. Having separate models as these would be useful for finding such insights that might improve the process efficiency better. Furthermore, there might also be some research possibilities for using the worker position data for other types of decision-making in manufacturing rather than worker capacity allocation to tasks. For example, one can investigate the decisions regarding the products, such as how many products to manufacture in a day or when to start the activities for each new product. Given that indoor localisation sensors are becoming commonplace in manufacturing environments due to their affordability and recent advances in this technology, it is of interest to investigate the potential of worker position data to inform and improve the manufacturing processes. Another possibility for future research can involve the implementation of the capacity adjustment strategies provided in this paper as suggestions to the workers in the considered assembly line. After the suggestions have been placed, new position data can be collected from workers to observe how they change their work practices and to see how much improvement has been achieved. The new data can then be also used for deriving process models and building a discrete event simulation for decision support using our methodology to provide further suggestions for process enhancement.

## Figures and Tables

**Figure 1 sensors-23-04928-f001:**
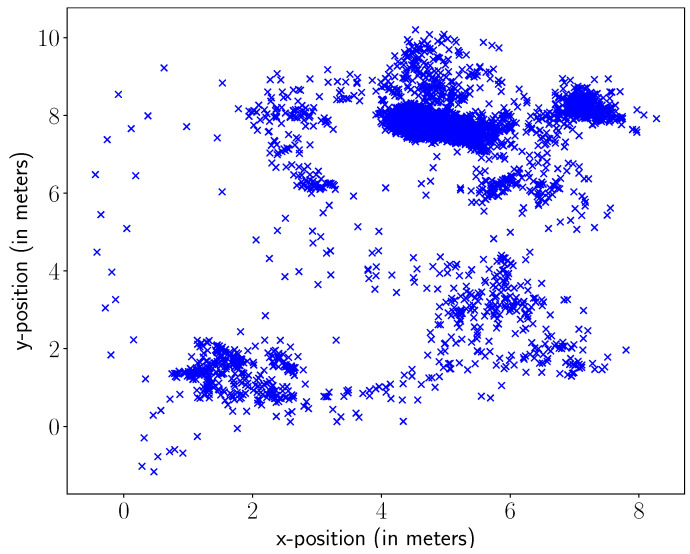
Position Data of Worker 1.

**Figure 2 sensors-23-04928-f002:**
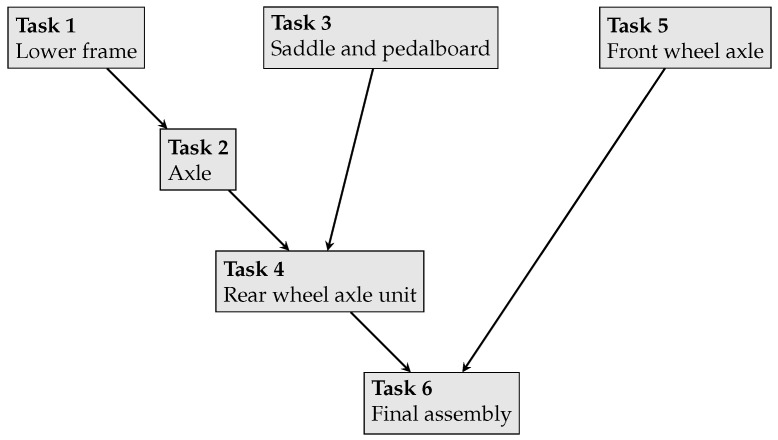
The manufacturing tasks of assembling tricycles.

**Figure 3 sensors-23-04928-f003:**
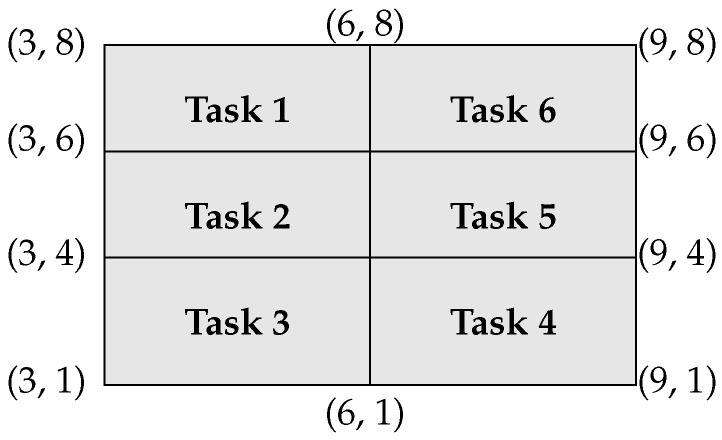
The locations of regions designated for tasks. (Region for Task 1 is defined as {(x,y)|3≤x<6,6<y≤8}, for Task 2 as {(x,y)|3≤x<6,4<y≤6}, for Task 3 as {(x,y)|3≤x<6,1<y≤4}, for Task 4 as {(x,y)|6≤x<9,1<y≤4}, for Task 5 as {(x,y)|6≤x<9,4<y≤6}, and for Task 6 as {(x,y)|6≤x<9,6<y≤8}.)

**Figure 4 sensors-23-04928-f004:**
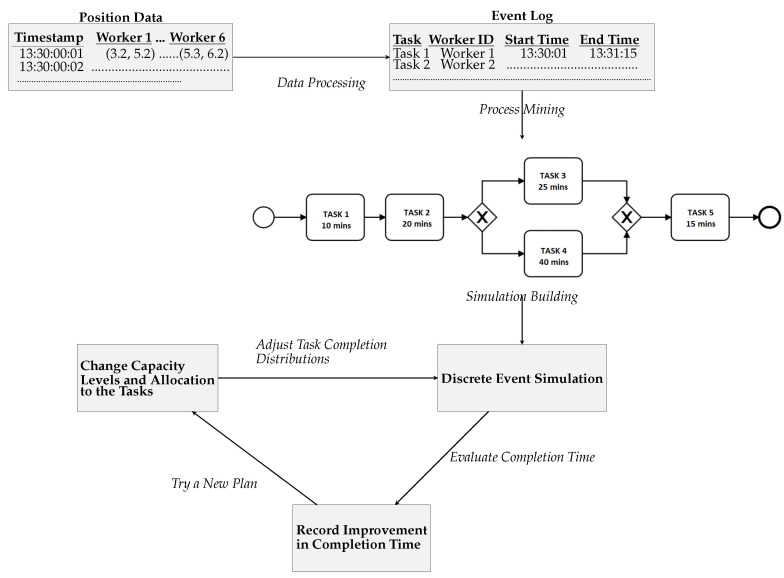
Methodology: using worker position data for human-driven process model discovery and decision support with discrete event simulation.

**Figure 5 sensors-23-04928-f005:**
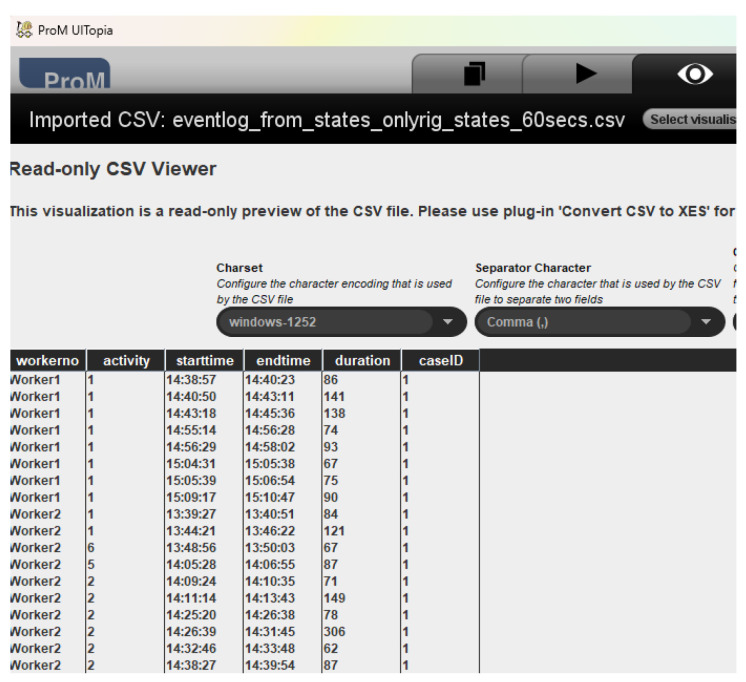
Inspecting the Event Log provided in CSV format in ProM.

**Figure 6 sensors-23-04928-f006:**
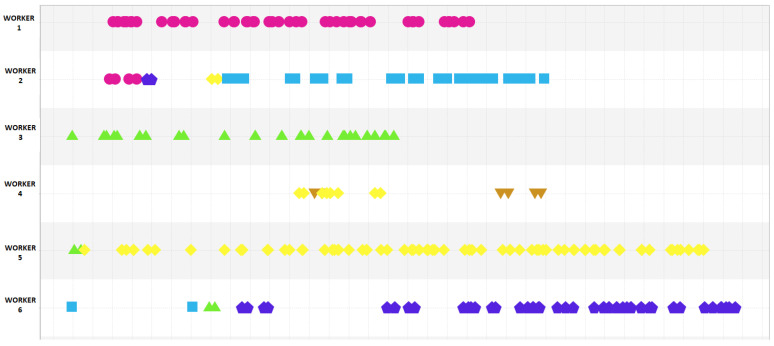
Dot chart of manufacturing tasks performed by 6 workers. (The points mark the start time of the activities (Task 1: Circle (Pink), Task 2: Square (Blue), Task 3: Up-pointing Triangle (Green), Task 4: Down-pointing Triangle (Orange), Task 5: Diamond (Yellow), Task 6: Pentagon (Purple)).

**Figure 7 sensors-23-04928-f007:**
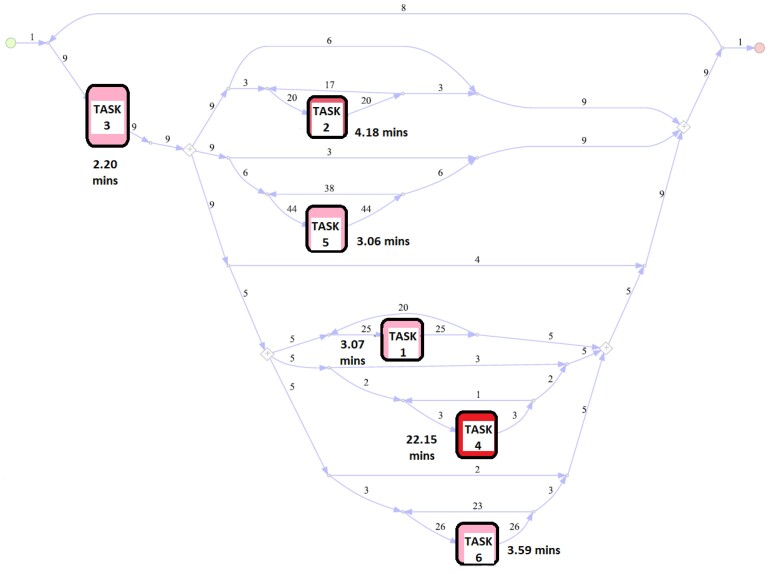
The process model with mean sojourn times (in minutes) of tasks and flow frequencies (mean sojourn times are (3.07,4.18,2.20,22.15,3.06,3.59)).

**Figure 8 sensors-23-04928-f008:**
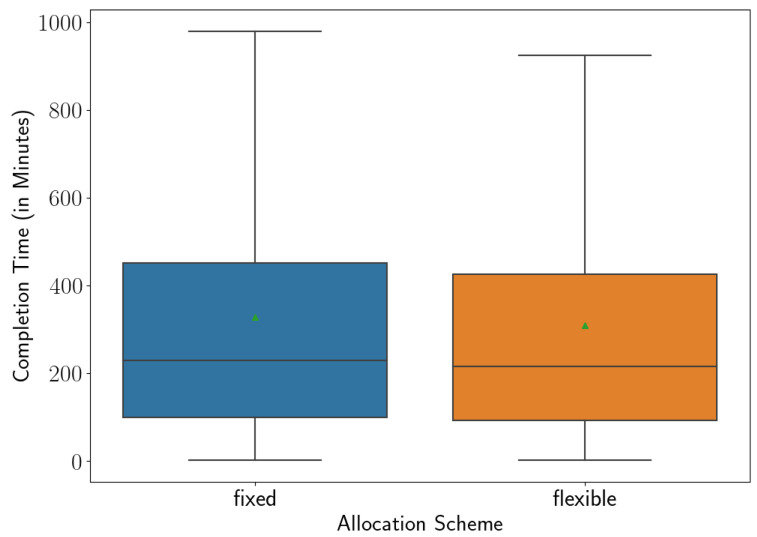
The completion time under the fixed and flexible allocations (green triangles mark the means. With 95% Confidence, the means of completion times under the fixed and flexible allocations are in the intervals (328.5, 328.7) and (309.7, 309.9), respectively.)

**Figure 9 sensors-23-04928-f009:**
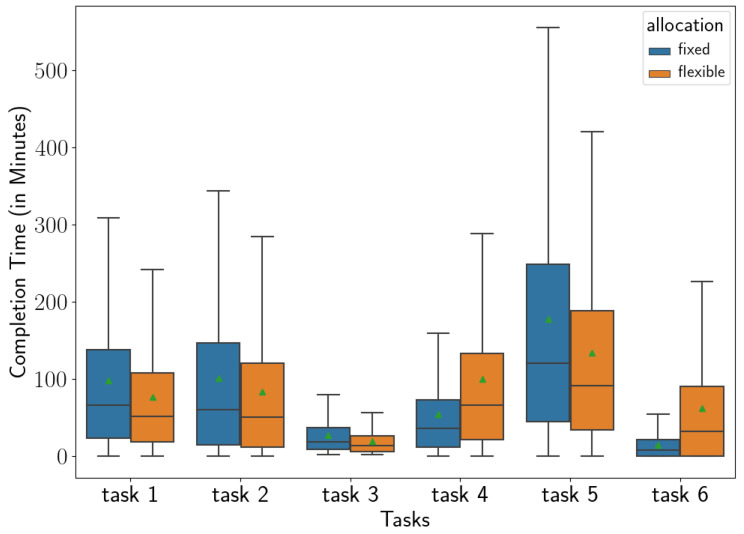
The total time spent in manufacturing tasks in the fixed versus flexible allocations (green triangles mark the means. $R^{\mathrm{fixed}}=\left(1,1,1,1,1,1\right)$ and $R^{\mathrm{data}}=\left(1.28,1.21,1.41,0.55,1.32,0.24\right)$).

**Figure 10 sensors-23-04928-f010:**
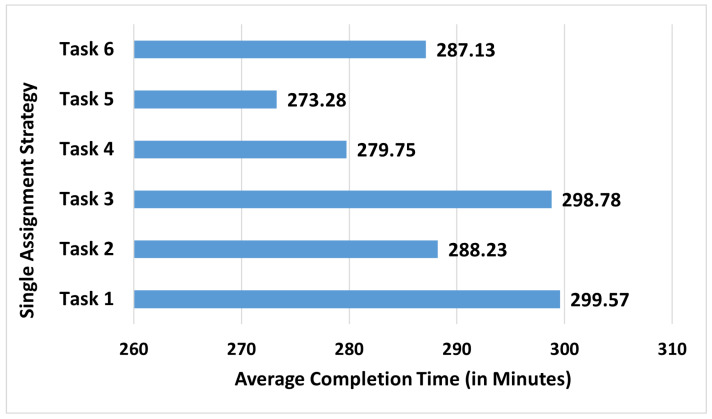
The average completion time (in minutes) after adding the capacity of the additional worker to the worker-driven flexible allocation through single-task assignment (The standard errors vary between 1.2 and 1.3 min).

**Figure 11 sensors-23-04928-f011:**
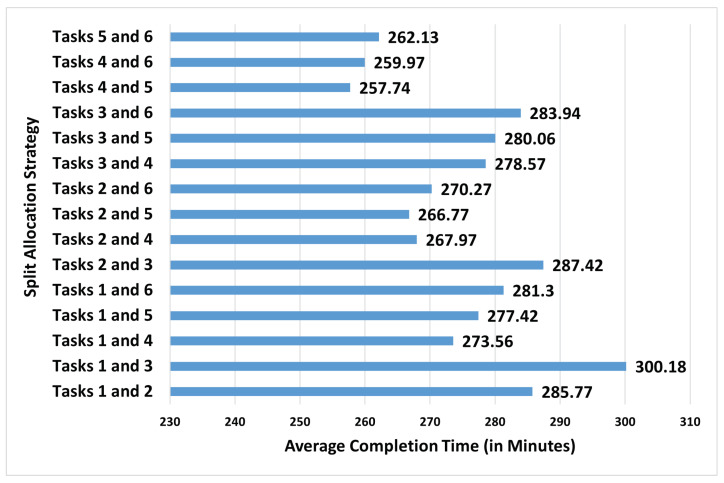
The average completion time (in minutes) after adding the capacity of the additional worker to the worker-driven flexible allocation through two-tasks assignment (The standard errors vary between 1.1 and 1.3 min).

**Table 1 sensors-23-04928-t001:** Comparison to previous studies using indoor localisation sensor data for decision support in manufacturing.

Study	Sensor Technology	Tracked Objects	Purpose	Technique of Analysis
[14]	RFID	products	job scheduling in machines	simulation
[15]	RFID	products	re-design of workflow	simulation
[16]	RFID	products	production scheduling	simulation
[17]	RFID	products	production scheduling	multi objective optimisation
[18]	Fixture	workstations	fault detection	quadratic programming
[19]	UWB	products	lead time prediction	semantic enriching
[20]	Lidar and UWB	workers	safety monitoring	supervised machine learning
[21]	UWB	products	bottleneck identification	process mining and value stream mapping
[22]	UWB	workers	safety monitoring	supervised machine learning
[23]	UWB	products	bottleneck identification	value stream mapping
Our paper	UWB	workers	worker capacity allocation	process mining and simulation

**Table 2 sensors-23-04928-t002:** The average completion times (in minutes) under shifting capacity between tasks based on the flexible allocation.

Adjustment	Shift 5%	Shift 10%	Shift 20%	Shift 50%
From Task 1 to 2	308.4 ± 1.4	307.6 ± 1.4	308.1 ± 1.4	332.7 ± 1.5
From Task 1 to 3	313.4 ± 1.4	314.4 ± 1.4	317.8 ± 1.4	347.2 ± 1.5
From Task 1 to 4	302.2 ± 1.3	299.5 ± 1.3	298.3 ± 1.3	326.7 ± 1.4
From Task 1 to 5	307.0 ± 1.4	305.0 ± 1.3	303.3 ± 1.3	323.9 ± 1.4
From Task 1 to 6	303.8 ± 1.3	301.1 ± 1.3	301.6 ± 1.3	333.2 ± 1.4
From Task 2 to 1	311.2 ± 1.4	313.7 ± 1.4	320.8 ± 1.4	370.2 ± 1.7
From Task 2 to 3	311.3 ± 1.4	314.1 ± 1.4	321.6 ± 1.4	370.4 ± 1.6
From Task 2 to 4	307.3 ± 1.3	306.3 ± 1.3	309.0 ± 1.4	354.0 ± 1.6
From Task 2 to 5	307.8 ± 1.4	307.8 ± 1.4	310.7 ± 1.4	352.0 ± 1.6
From Task 2 to 6	303.5 ± 1.3	302.7 ± 1.3	307.0 ± 1.3	356.7 ± 1.6
From Task 3 to 1	308.4 ± 1.4	308.3 ± 1.4	308.9 ± 1.4	320.0 ± 1.4
From Task 3 to 2	308.8 ± 1.3	307.1 ± 1.3	305.1 ± 1.3	310.8 ± 1.4
From Task 3 to 4	302.5 ± 1.3	298.6 ± 1.3	294.5 ± 1.3	300.6 ± 1.3
From Task 3 to 5	306.2 ± 1.3	303.1 ± 1.3	298.7 ± 1.3	299.1 ± 1.3
From Task 3 to 6	302.4 ± 1.3	298.7 ± 1.3	296.8 ± 1.3	308.1 ± 1.3
From Task 4 to 1	313.3 ± 1.4	316.6 ± 1.4	325.4 ± 1.4	384.4 ± 1.7
From Task 4 to 2	311.0 ± 1.4	313.6 ± 1.4	321.1 ± 1.4	376.9 ± 1.7
From Task 4 to 3	312.2 ± 1.4	315.7 ± 1.4	324.7 ± 1.4	383.2 ± 1.7
From Task 4 to 5	309.8 ± 1.4	311.9 ± 1.4	318.3 ± 1.4	371.1 ± 1.6
From Task 4 to 6	307.7 ± 1.4	308.5 ± 1.4	314.2 ± 1.4	371.6 ± 1.6
From Task 5 to 1	314.8 ± 1.4	319.3 ± 1.4	331.5 ± 1.5	411.7 ± 1.8
From Task 5 to 2	309.5 ± 1.3	312.5 ± 1.4	322.1 ± 1.4	397.4 ± 1.7
From Task 5 to 3	313.2 ± 1.4	318.1 ± 1.4	330.6 ± 1.4	409.6 ± 1.8
From Task 5 to 4	307.9 ± 1.3	308.8 ± 1.4	316.7 ± 1.4	394.0 ± 1.7
From Task 5 to 6	305.1 ± 1.3	306.2 ± 1.3	315.9 ± 1.4	396.7 ± 1.7
From Task 1 to 4 and From Task 3 to 6			287 ± 1.4	

## Data Availability

The data (UWB position data and the event log in CSV format) and C++ source codes of the simulation can be found in the data repository of Edinburgh Napier University. See https://doi.org/10.17869/enu.2023.3100035.

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
