# Peer review of "Using Worker Position Data for Human-Driven Decision Support in Labour-Intensive Manufacturing"

_sensors, 2023, doi:10.3390/s23104928_

Round 1

Reviewer 1 Report

Dear Authors, please find my comments below regarding the paper: Using Worker Position Data for Human-Driven Decision
Support in Labour-intensive Manufacturing

C1: Abstract is good and concise.
C2: Introduction is a bit long, i think the same information could be delivered in shorter format too.
C3: There are more categories in IPS technologies, than described in the manuscript. RFID, IMU, ZigBee, WiFi, Ultrasound, BT, LWB, IR... I know that some of this can be fitted in UWB, but it would be better to better dive into these terms separately. Also you submitted the paper to Sensors, please emphasize the connection between the journal and the topic.
C4: There are key researches cited in the paper - please add relevant author names. (e.g. [20] -> "the work of Tran et al. served as a relevant basis for our study...")
C5: Chapter 2 is also bloated/wordy - please try to make it more compact.
C6: Fig 1 is not too clear -> please increase text size, remove title, and if possible render the figure in a more professional tool (not Excel). Use vectors for all figures!
C7: The first paragraph of 3.2.1 is a good example again for lengthy writing.
"UWB tags measure at an interval of 100 ms, but the position data collected from workers may not be synchronised, and there may be gaps in the data at times when the signal is obstructed by metal objects near the workers. To align the data from all workers,  we use seconds as the time unit since we don’t expect the workers’ positions to change significantly within a second. If multiple position data points are obtained from a worker  within the same second, we take the average of those points as the worker’s position for that second, which also serves as a smoothing function to mitigate measurement errors. However, if there are instances where no position data is collected for a worker, we substitute the missing data with the worker’s last known position."
Shortened example:
"UWB tags measure with 100 ms time step, but the position of the workers may not be synchronised, and signal obstruction (e.g. by metal objects) cause timely gaps. Practically a second base evaluation of the data is enough for the positioning. If multiple positions are obtained within a second, an average is calculated, which also serves as a smoothing function to mitigate measurement errors. If no position data is collected, we substitute the missing data with the worker’s last known position."
C8: Please separate all equations from the text.
C9: Please improve readability of Fig 9 by increasing the number's size. Also highlight of milliseconds is not necessary. (Are they?)
C10: Fig 10 legend is incorrect (fixed, data -> flexible?), also please evaluate statistical significance of difference between the two results. The Y axis unit is missing.
C11: Fig 11 is also incorrect (legend), the style and format is not aligned with Fig 10, and Y axis title and unit is missing. Harmonize style with Fig10 and please use a less apparently Excel-style format (even if using Excel).
C12: The summary is also long -> there is a need to clearly formulate the novelties and highlight them ina concise format at the end of the paper.
C13: I would consider to add Appendix parts to the methodology descriptions in Ch3.

The quality of the text is ok, from the aspect of English. Only the length should be shortened.

Reviewer 2 Report

The authors demonstrate the generation of data-driven process models but did not report how this output could be used to create simulation models and decision support for process improvement. The main goal of this paper is to show the potential of worker position data, for providing human-driven decision support to manufacturers via the application of process mining algorithms and discrete event simulation tool.

The paper is interesting and is well written. However, some major changes are needed:

- In scientific-technical documents, the first-person plural ("we") must not be used. Please use passive or third person voice.

- The font size in Fig 6,7, 9 are very small. Please, increase the size of these fonts.

- The conclusions are not clear. Please, check this.

- The state of the art is very focused on the human factor and its relationship with Labor-Intensive Manufacturing. However, I think it is necessary, in order to improve the impact of the publication, to include references to sustainability aspects.

Goel P., et ál. Sustainable Green Human Resource Management Practices in Educational Institutions: An Interpretive Structural Modelling and Analytic Hierarchy Process Approach (2022) Sustainability (Switzerland), 14 (19), art. no. 12853, DOI: 10.3390/su141912853

Eliminate first person of plural ("we"). Use passive voice.

Reviewer 3 Report

Attached

Round 2

Reviewer 1 Report

Thank you for the corrections.

Reviewer 2 Report

The authors have modified the article according to the reviewer´s comments. For this, the paper have improved in terms of scientific soundness and quality of presentation.

Reviewer 3 Report

COMMENTS CARRIEDOUT IN THE REVISED VERSION..